# PocketSR: The Super-Resolution Expert in Your Pocket Mobiles

**Haoze Sun[1]**[*]**, Linfeng Jiang[2]**[*]**, Fan Li[2], Renjing Pei[2], Zhixin Wang[2], Yong Guo[2],**
**Jiaqi Xu[2], Haoyu Chen[4], Jin Han[2], Fenglong Song[2], Yujiu Yang[1]**[✉]**, Wenbo Li[3]**[✉]
[1]Tsinghua University    [2]Huawei    [3]Joy Future Academy    [4]HKUST (GZ)
shz22@tsinghua.org.cn   yang.yujiu@sz.tsinghua.edu.cn   fenglinglwb@gmail.com

## Abstract

Real-world image super-resolution (RealSR) aims to enhance the visual quality of in-the-wild images, such as those captured by mobile phones. While existing methods leveraging large generative models demonstrate impressive results, the high computational cost and latency make them impractical for edge deployment. In this paper, we introduce PocketSR, an ultra-lightweight, single-step model that brings generative modeling capabilities to RealSR while maintaining high fidelity. To achieve this, we design LiteED, a highly efficient alternative to the original computationally intensive VAE in SD, reducing parameters by 97.5% while preserving high-quality encoding and decoding. Additionally, we propose online annealing pruning for the U-Net, which progressively shifts generative priors from heavy modules to lightweight counterparts, ensuring effective knowledge transfer and further optimizing efficiency. To mitigate the loss of prior knowledge during pruning, we incorporate a multi-layer feature distillation loss. Through an in-depth analysis of each design component, we provide valuable insights for future research. PocketSR, with a model size of 146M parameters, processes 4K images in just 0.8 seconds, achieving a remarkable speedup over previous methods. Notably, it delivers performance on par with state-of-the-art single-step and even multi-step RealSR models, making it a highly practical solution for edge-device applications.

## 1 Introduction

Real-world image super-resolution (RealSR) [1, 2, 3, 4, 5, 6, 7] is a fundamental task in computer vision that reconstructs high-quality images from low-quality inputs. With applications spanning smartphone photography, medical imaging, and remote sensing, RealSR has driven extensive research interest and development. Recent breakthroughs in generative modeling—particularly diffusion models [8, 9, 10, 11, 12, 13, 14]—have opened new frontiers in SR by leveraging powerful generative priors to recover intricate textures and realistic structures, significantly enhancing image quality.

Scaling up text-to-image (T2I) diffusion models (*e.g.*, from SD1.5 [10] to SD3 [15] or FLUX [16]) has been shown to markedly improve SR performance [17]. Additionally, integrating multimodal large language models to generate detailed and accurate image descriptions further unlocks these models' generative potential [18, 19, 20, 21, 22]. However, the substantial computational cost and slow inference speed of such large-scale models limit their practical deployment. Consequently, research efforts have shifted toward efficient SR diffusion methods, focusing on lightweight architectures [23] and reduced sampling steps [24, 25, 26, 27, 28]. Yet, deploying these models on resource-constrained edge devices remains a formidable challenge.

Given that low-resolution (LR) inputs provide a rich and detailed prior—far more informative than the sparse textual cues in T2I generation—it is possible to achieve competitive SR performance with

---

[*]Equal Contribution    [✉] Corresponding Author

39th Conference on Neural Information Processing Systems (NeurIPS 2025).

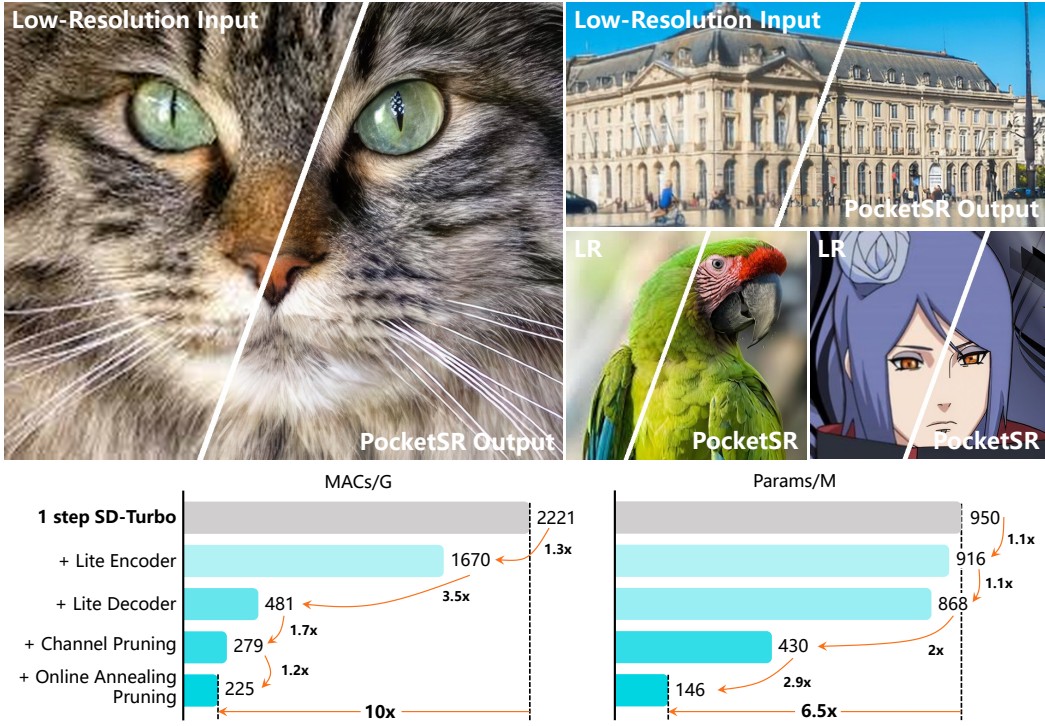

Figure 1: Visualization of the real-world image super-resolution results and efficiency of our proposed method. To enable the practical application of diffusion-based SR models, we introduce PocketSR, an ultra-lightweight, single-step solution. The top visual examples demonstrate that PocketSR achieves high-quality super-resolution across diverse scenes, preserving fine details and textures. Best viewed zoomed in. The bottom section highlights the significant reduction in parameters ($10\times$) and computational cost ($6.5\times$), allowing our model to process 4K images in just 0.8 seconds—dramatically outpacing existing methods.

a significantly compressed model. Motivated by this insight, we conduct a thorough analysis of the core components of diffusion models, including the variational autoencoder (VAE) and the U-Net architecture, and propose tailored strategies to maintain high-fidelity and realistic generation while substantially reducing the model size. Our approach is built upon three key innovations:

**Lite Encoder & Decoder (LiteED).** We introduce a highly compressed encoder-decoder design featuring an ultra-compact encoder for efficient LR feature extraction and conditioning, coupled with a tiny decoder for high-quality reconstruction. To mitigate potential losses in representational capacity, we incorporate a Dual-Path Feature Injection mechanism that enriches U-Net inputs with additional high-dimensional feature channels and Adaptive Skip Connections that retain critical information. The resulting LiteED contains only 2M parameters, drastically improving model efficiency while preserving image quality.

**Online Annealing Pruning for U-Net.** Recognizing the extensive generative priors embedded within U-Net architectures, we go beyond straightforward channel pruning [23] by introducing an online annealing pruning strategy. In this approach, lightweight modules are gradually integrated alongside existing components (*e.g.*, residual blocks, self-attention layers, and feed-forward networks), while the contributions of original modules are progressively annealed to zero. This smooth transition ensures a stable knowledge transfer to the pruned architecture. Additionally, we conduct a comprehensive ablation study to determine optimal pruning positions, ensuring that our lightweight model retains strong generative capabilities.

**Multi-layer Feature Distillation.** Prior studies [23, 29] have demonstrated that feature-space distillation offers a more stable optimization process compared to distillation in the image domain. Motivated by this observation, we adopt a multi-layer, multi-scale distillation scheme to facilitate more reliable knowledge transfer. When combined with our online annealing pruning strategy, this approach substantially reduces computational cost while effectively preserving generative priors.

Through these architectural optimizations, we further integrate adversarial training to develop PocketSR, an ultra-lightweight, one-step diffusion-based SR model with only 146M parameters—just 10.4% of the size of StableSR [3] and 8.2% of OSEDiff [27]. Despite its compactness, PocketSR achieves performance on par with state-of-the-art approaches while significantly reducing computational complexity and inference time, as illustrated in Sect. 4.2. Notably, it processes a 4K image in just 0.8 seconds—7 times faster than OSEDiff [27]. Our findings demonstrate that with carefully designed modifications, diffusion-based SR models can be both lightweight and powerful, paving the way for practical deployment in real-world applications.

## 2 Related Work

### 2.1 Real-world Image Super-Resolution

Real-world image super-resolution (SR) aims to recover realistic details from degraded inputs while maintaining overall fidelity. Early image super-resolution (ISR) methods [30, 31, 32, 33] often converge to the statistical mean of plausible solutions, resulting in overly smoothed outputs and loss of fine details in real-world scenarios. To overcome this, several works [34, 2, 1, 35, 36, 37] have explored generative adversarial networks (GANs) for texture enhancement. However, due to inherent limitations such as mode collapse and training instability [38, 39], GAN-based methods still struggle to produce realistic textures. More recently, diffusion models [8, 9, 10, 11, 12, 40] have shown strong generative capabilities in synthesizing fine-grained details and realistic textures, making them a promising alternative for real-world SR.

### 2.2 Diffusion-based ISR

Recent advances in diffusion models, especially in text-to-image (T2I) synthesis (e.g., SD3 [15], FLUX [16]), have led to the development of pre-trained diffusion-based SR methods such as StableSR [3], DiffBIR [4], PASD [5], CoSeR [41], SeeSR [19], and SUPIR [18]. These approaches benefit from the strong image priors of T2I models, achieving impressive results. However, their reliance on multi-step inference introduces high latency and substantial computational cost.

To address this, recent works aim to reduce inference time by distilling multi-step models into one- or few-step variants. ResShift [6] accelerates inference by learning residual transitions from LR to HR images via a Markov chain. Building on this, Wang et al.[24] condense multi-step capabilities into single-step networks. OSEDiff[27] further leverages VSD [42] to incorporate T2I knowledge efficiently. PiSA-SR [43] decouples structure restoration and texture enhancement using dual LoRA sets. Despite these advances, single-step SR methods still remain computationally heavier than traditional GAN-based models, limiting their deployment on edge devices and highlighting the need for better efficiency–quality trade-offs.

### 2.3 Efficient Diffusion Models

Recent efforts on efficient diffusion models [44, 45, 46, 47, 48, 49, 29] focus on architectural optimization to reduce redundancy in large-scale models. SnapFusion [49] disentangles the contributions of individual modules to balance efficiency and accuracy. MobileDiff [47] improves efficiency by relocating Transformer blocks to lower-resolution stages. SnapGen [29] cuts computation and model size by removing high-resolution attention and replacing standard convolutions with depthwise separable ones. AdcSR [23] further explores one-step SR model efficiency. In Sect. 4.3, we compare our design with AdcSR and show that our approach achieves better performance with higher compression.

## 3 Method

Driven by deployment considerations, our PocketSR framework utilizes SD-Turbo[2] as the backbone, which achieves extreme model compression through a two-stage training pipeline. In the first stage, we replace the original Stable Diffusion's (SD) variational autoencoder with the Lite Encoder-Decoder (LiteED), with all parameters set to be trainable. In the second stage, we freeze LiteED and apply our pruning strategies to the U-Net, gradually removing redundant components.

---

[2]https://huggingface.co/stabilityai/sd-turbo

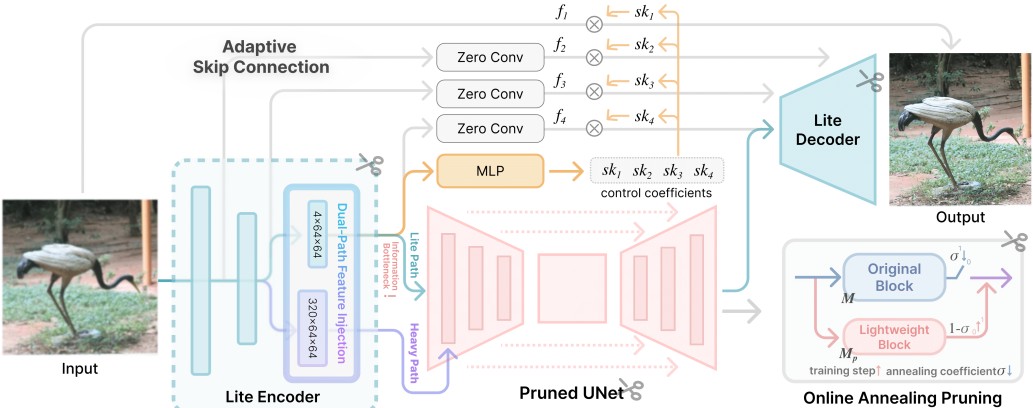

Figure 2: Overview of PocketSR framework. We replace the original Stable Diffusion variational autoencoder with LiteED, and apply online annealing pruning and multi-layer feature distillation strategies to the diffusion U-Net, effectively reducing model parameters and computational complexity while maintaining excellent super-resolution performance.

## 3.1 Lite Encoder & Decoder

We propose LiteED, an ultra-lightweight encoder-decoder architecture designed for edge-side super-resolution. To reduce complexity, we simplify the SD encoder to several convolutional layers, while maintaining effective feature extraction. Additionally, we replace the original decoder with a lightweight alternative that offers substantial improvements in efficiency while preserving comparable image fidelity (detailed in the supplementary materials). Notably, the decoder in LiteED is modular and can be substituted with a more powerful variant to enhance reconstruction quality, albeit at the cost of efficiency. The decoder is initialized from an open-source model[3], while the encoder is randomly initialized.

To supplement the ultra-light encoder and enrich the representational capacity, we propose an adaptive skip connection mechanism in LiteED, as depicted in Figure 2. Specifically, four control coefficients are generated from the encoder output via an MLP to modulate the multi-scale skip connections. This adaptive mechanism allows the model to selectively integrate input features during decoding, effectively mitigating the loss of information due to encoder compression. Additionally, to stabilize training, we incorporate zero-convolution into the skip connections.

**Dual-path Feature Injection.** We identify an information bottleneck between the original SD encoder and the U-Net. For a $512 \times 512$ image, the output feature size of the SD encoder's last ResBlock is $[N, 512, 64, 64]$, but it is compressed to $[N, 4, 64, 64]$ in the final convolutional layer. This 128× reduction in feature dimensionality may lead to substantial information loss, negatively impacting super-resolution performance. To address this issue and improve the fidelity of the result, we propose a dual-path feature injection mechanism. Specifically, in our encoder, an additional high-dimensional feature of size $[N, 320, 64, 64]$ is extracted after the second convolutional layer of LiteED and injected into the U-Net following the initial block. This strategy enhances the information flow, allowing for more robust feature extraction. As for feature injection, we employ cross normalization [50], which has been shown to facilitate fast and stable convergence.

Despite its lightweight design, LiteED proves to be more effective in practice than networks with several times higher computational complexity. Moreover, the decoder in LiteED can be replaced with a larger-capacity network, which leads to better performance as demonstrated in our experiments.

## 3.2 U-Net Pruning

### 3.2.1 Online Annealing Pruning

Pruning is a straight-forward and effective model compression technique, which is widely used in efficient image generation [45, 48, 47, 49, 29, 51, 52]. In this paper, we propose online annealing

---

[3]https://github.com/madebyollin/taesd

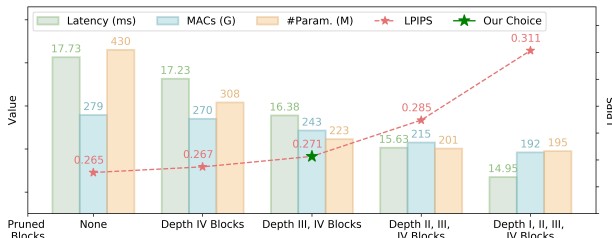

Figure 3: Analysis of the impact of pruning residual blocks at different depths using the widely adopted RealSR [53] test set, with performance measured by the LPIPS [54] metric. The inference resolution is $512 \times 512$, and we report the maximum inference speed on an A100 GPU.

Table 1: Comparison of the computational efficiency of the SD network and its modules before and after our streamlining. We highlight the computational efficiency of the original SD and the improvements brought by PocketSR.

| Network | Computational Efficiency | | |
|---|---|---|---|
| | Time (ms) | MACs (G) | #Param. (M) |
| Lite Encoder | ↓99% 0.2 | ↓99% 8.1 | ↓98% 0.8 |
| SD Encoder | 22.6 | 559.5 | 34.2 |
| Pruned U-Net | ↓53% 12.5 | ↓64% 146.1 | ↓83% 144.2 |
| SD U-Net | 26.7 | 401.9 | 866 |
| Lite Decoder | ↓90% 3.0 | ↓94% 70.7 | ↓98% 1.2 |
| SD Decoder | 30.3 | 1259.7 | 49.4 |
| PocketSR | ↓80% 15.7 | ↓90% 224.9 | ↓85% 146.2 |
| Original SD | 79.7 | 2221.2 | 949.6 |

pruning, a stable and efficient pruning strategy specifically designed for SR. Existing Diffusion U-Net pruning methods typically simply discard [47, 49] or replace pruned modules [48, 29], which leads to a significant loss of prior information. Our pruning strategy preserves the prior information through online knowledge transfer. As shown in Figure 2, we connect the original module $M$ in parallel with a lightweight replacement module $M_P$. During the training process, we continuously increase the contribution of the lightweight module, while gradually annealing the contribution of the original module to zero:

$$\mathbf{y} = \sigma \cdot M(\mathbf{x}) + (1 - \sigma) \cdot M_P(\mathbf{x}), \tag{1}$$

where $\mathbf{x}$ and $\mathbf{y}$ represent the module input and output, respectively. The annealing coefficient is defined as $\sigma = \min\left(0, (T - t)/T\right)$, where $T$ is the total number of annealing steps, and $t$ denotes the current training step. The parameters of the original module are frozen during training. Once training is complete, the original module is replaced with the lightweight module. The pruned modules are permanently removed at inference time, yielding a highly compact and efficient model. Our experiments show that this online pruning approach can better preserve the diffusion prior and generate more realistic textures (see Sect. 4.3).

### 3.2.2 Pruning Implementation

Diffusion U-Net comprises four computationally expensive module types: residual blocks, cross-attention layers, self-attention layers, and feed-forward networks (FFNs). To reduce the number of parameters and computational cost while preserving performance, we replace each module with a carefully chosen lightweight counterpart. Lightweight layers (e.g., normalization, activation) remain unchanged. For residual blocks, we replace all convolutions with depthwise separable convolutions [55], significantly reducing the number of parameters. Since text input is entirely discarded, cross-attention layers are replaced with MLPs of two linear layers. Self-attention layers are approximated using linear attention [56] to reduce computational cost. Finally, the hidden dimension of the FFN is reduced to one-fourth of its original size.

We also observe that the pruning location has a significant impact on model performance, with its effect varying according to the module's depth within the network. Specifically, *shallower modules exert a greater influence on super-resolution quality, while deeper blocks can be pruned with minimal impact.* Taking SD's U-Net as an example, we label the depths as {I, II, III, and IV}, from shallow to deep. Figure 3 illustrates the trade-off between performance and efficiency when pruning residual blocks at different depths. Our results show that pruning at deeper levels (III and IV) significantly reduces parameter count and computational cost while having negligible impact on performance. While pruning shallower modules yields greater efficiency gains, it leads to a more noticeable drop in final performance.

We interpret this phenomenon as deeper layers in the U-Net of a generative diffusion model primarily process high-level information [57, 58], such as layout and style, which contributes less to the SR task. This characteristic makes a free lunch for pruning. Our findings extend beyond residual blocks to other network modules, *e.g.*, self-attention, cross-attention, and FFNs, indicating a general pruning strategy for SR networks. Balancing efficiency and effectiveness, we prune residual blocks and FFNs at depths III and IV, while self-attention layers are pruned at depth IV. Additionally, we prune

all cross-attention layers, as their impact on super-resolution quality is minimal. Detailed ablation experiments on all network module pruning locations are provided in the supplementary material.

### 3.2.3 Multi-layer Feature Distillation

To better preserve generative priors during pruning, we introduce a feature-level knowledge distillation strategy. Due to the architectural mismatch between the models before and after pruning, single-layer distillation may result in training instability. Instead, we adopt a multi-layer global distillation scheme to enhance robustness and enable stable knowledge transfer:

$$\mathcal{L}_{\text{distill}} = \mathbb{E}\left[\sum_l \left\| f_{\text{pre-pruning}}^l - \phi\left(f_{\text{post-pruning}}^l\right) \right\|_2^2\right], \tag{2}$$

where $f_{\text{pre-pruning}}^l$ and $f_{\text{post-pruning}}^l$ denote the feature representations from the $l$-th layer of the models before and after pruning, respectively. Following [59], we implement the mapping function $\phi(\cdot)$ as a lightweight, trainable projection module composed of a single convolutional layer.

### 3.3 Training Details

Beyond the aforementioned lightweight optimizations, we further investigate the impact of channel reduction on model performance, which is compatible with our online annealing pruning strategy. Notably, direct channel reduction can substantially weaken the model's learned priors. However, this adverse effect is markedly mitigated when integrated with the proposed multi-layer feature distillation strategy. Comprehensive experimental results are provided in the supplementary materials. To strike a balance between efficiency and performance, we reduce the channel width in the U-Net to 70% of its original size. Table 1 presents the final latency, computational cost, and parameter count of our model in comparison to the original SD-Turbo.

Building on recent progress in one-step diffusion-based generation [60, 61], we introduce adversarial loss during training to improve texture fidelity. In the first stage, the model is trained using a combination of MSE loss, LPIPS [62], and adversarial loss. In the second stage, we incorporate the multi-layer feature distillation loss to retain knowledge from the full-capacity model during pruning.

In the first training phase, we train the unpruned SD U-Net equipped with LiteED for 80,000 steps. In the second phase, we first apply channel pruning over 80,000 steps, followed by module-wise online annealing pruning for an additional 8,000 steps. The total number of annealing steps is set to $T = 8000$. A fixed batch size of $64$ is used throughout the entire training process. We employ the AdamW optimizer with a learning rate of $1 \times 10^{-4}$, and the timestep is fixed at $t = 999$ for one-step diffusion. Additionally, the original text embedding is replaced with a learnable embedding vector.

## 4 Experiments

### 4.1 Experimental Settings

The training dataset comprises approximately 500K high-quality images from LSDIR [63] and 10K images from FFHQ [64]. During training, images are randomly cropped into $512 \times 512$ patches. Low-quality counterparts are synthesized using the widely adopted degradation pipeline from Real-ESRGAN [1]. For evaluation, we follow the protocols in [23, 27] and report results on the DRealSR [65] and RealSR [53] benchmarks.

We compare the proposed PocketSR with four state-of-the-art multi-step methods—StableSR [3], DiffBIR [4], SeeSR [19], and ResShift [6]—as well as three leading single-step methods: SinSR [24], AdcSR [23], and OSEDiff [27]. We also compare our model with GAN-based SR methods in the supplementary material. All models are evaluated using a comprehensive set of metrics, including perceptual similarity metrics (LPIPS [62], DISTS [66]), fidelity metrics (PSNR, SSIM [67]), and no-reference quality metrics (NIQE [68], MUSIQ [69]).

### 4.2 Comparison with State-of-the-Arts

**Quantitative and Efficiency Comparison.** Table 2 reports quantitative results on RealSR [53] and DRealSR [65], with efficiency metrics listed in the last five rows. The results show that our

Table 2: Quantitative and efficiency comparisons with state-of-the-art diffusion-based methods on real-world datasets are presented. The best results for each metric are highlighted in bold. PocketSR delivers real-time inference at over 60 FPS on an A100 GPU for $512 \times 512$ inputs, while maintaining competitive quantitative performance.

| Datasets | Metrics | StableSR [3] | DiffBIR [4] | SeeSR [19] | ResShift [6] | SinSR [24] | OSEDiff [27] | AdcSR [23] | **PocketSR** |
|---|---|---|---|---|---|---|---|---|---|
| RealSR [53] | LPIPS↓ | 0.3018 | 0.3636 | 0.3009 | 0.3460 | 0.3188 | 0.2921 | 0.2885 | **0.2713** |
| | DISTS↓ | 0.2288 | 0.2312 | 0.2223 | 0.2498 | 0.2353 | 0.2128 | 0.2129 | **0.2094** |
| | PSNR↑ | 24.70 | 24.75 | 25.18 | **26.31** | 26.28 | 25.15 | 25.47 | 25.47 |
| | SSIM↑ | 0.7085 | 0.6567 | 0.7216 | **0.7421** | 0.7347 | 0.7341 | 0.7301 | 0.7330 |
| | NIQE↓ | 5.912 | 5.535 | 5.408 | 7.264 | 6.287 | 5.648 | 5.350 | **5.067** |
| | MUSIQ↑ | 65.78 | 64.98 | 69.77 | 58.43 | 60.80 | 69.09 | **69.90** | 67.07 |
| DRealSR [65] | LPIPS↓ | 0.3284 | 0.4557 | 0.3189 | 0.4006 | 0.3665 | 0.2968 | 0.3046 | **0.2962** |
| | DISTS↓ | 0.2269 | 0.2748 | 0.2315 | 0.2656 | 0.2485 | 0.2165 | 0.2200 | **0.2139** |
| | PSNR↑ | 28.03 | 26.71 | 28.17 | **28.46** | 28.36 | 27.92 | 28.10 | 28.05 |
| | SSIM↑ | 0.7536 | 0.6571 | 0.7691 | 0.7673 | 0.7515 | **0.7835** | 0.7726 | 0.7675 |
| | NIQE↓ | 6.524 | 6.312 | 6.397 | 8.125 | 6.991 | 6.490 | 6.450 | **5.809** |
| | MUSIQ↑ | 58.51 | 61.07 | 64.93 | 50.60 | 55.33 | 64.65 | **66.26** | 63.85 |
| Parameters (M) | | 1410 | 1717 | 2524 | **119** | 119 | 1775 | 456 | 146 |
| MACs (G) | | 79940 | 24234 | 65857 | 5491 | 2649 | 2265 | 496 | **225** |
| Sampling Steps | | 200 | 50 | 50 | 15 | 1 | 1 | 1 | 1 |
| Inference Time (s) | | 11.5 | 2.7 | 4.3 | 0.71 | 0.13 | 0.11 | 0.03 | **0.016** |
| FPS | | 0.09 | 0.37 | 0.23 | 1.4 | 7.7 | 9.1 | 33.3 | **62.5** |

method achieves strong super-resolution performance with excellent computational efficiency. First, the proposed single-step model, PocketSR, has only 146M parameters, achieves the lowest MACs among all methods, and processes a $512 \times 512$ image in 0.016 seconds on an A100 GPU—nearly twice as fast as AdcSR. This low latency and lightweight design make it ideal for real-time and edge deployment. PocketSR achieves an inference time of only 140ms on a newly released smartphone model from 2025, representing an over 80% reduction compared to the original, non-lightweight backbone. Second, PocketSR attains the best LPIPS, DISTS, and NIQE scores, demonstrating superior perceptual quality. Notably, on the DRealSR dataset, PocketSR surpasses the second-best method by **10%** in NIQE, indicating clearer texture restoration. Third, it delivers competitive fidelity, achieving PSNR and SSIM on par with AdcSR on RealSR and clearly outperforming multi-step models such as StableSR, DiffBIR, and SeeSR. Although ResShift and SinSR perform better than PocketSR in PSNR and SSIM, their texture are not as realistic as PocketSR.

**Qualitative Comparison.** Figure 4 shows qualitative comparisons, demonstrating that our method consistently maintains high fidelity and excels in detail reconstruction. In the first row, only DiffBIR, ResShift, SinSR, and PocketSR successfully recover the diagonal striped texture in the upper-right corner. However, the textures produced by DiffBIR and SinSR appear visually unrealistic. Among single-step methods, only PocketSR reconstructs an accurate and perceptually convincing pattern, highlighting its strength in preserving fine details. The second row further shows that PocketSR is the only method able to restore the building's structural details, reinforcing its high fidelity and detail recovery capabilities. Figure 1 showcases results across a broader range of natural scenes, including animals and animes. These results further verify that PocketSR not only achieves faithful reconstructions but also delivers rich, fine-grained textures across diverse visual contexts.

## 4.3 Ablation Study

We carefully analyze the effects of the proposed lite encoder and decoder, online annealing pruning, and multi-layer feature distillation, demonstrating an excellent trade-off between super-resolution quality and efficiency through extensive experiments. The experiments in this section follow the first-stage training in Section 3.3, where the unpruned SD U-Net is jointly trained with various encoder and decoder architectures for 80,000 steps. All settings share the same model and training configuration, differing only in encoder/decoder architecture. RealSR is used as the test dataset.

**Effect of Adaptive Skip Connection and Dual-path Feature Injection.** The proposed lite encoder simplifies the original SD encoder to improve efficiency, which inevitably introduces information

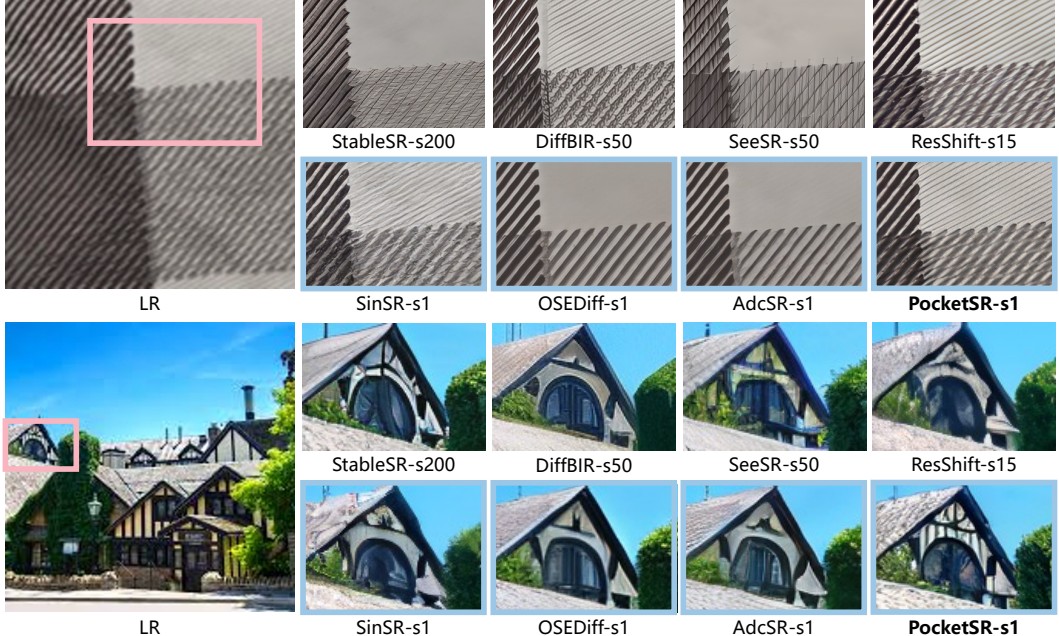

Figure 4: Qualitative results on real-world images. Single-step methods are highlighted for clarity. PocketSR delivers competitive performance, generating well-preserved structures and fine-grained textures, even when compared to multi-step models.

Table 3: We conducted ablation studies on the core architectural components of LiteED, including the Adaptive Skip Connection (ASC) and Dual-path Feature Injection (DFI). Additionally, we compared the proposed lightweight encoder with alternative designs to validate the effectiveness of our architecture. Finally, we replaced the decoder to assess the structural robustness of LiteED.

| Decoder | Encoder | PSNR↑ | SSIM↑ | LPIPS↓ | DISTS↓ | NIQE↓ | Time (ms) | MACs (G) | Param. (M) |
|---|---|---|---|---|---|---|---|---|---|
| **Lite Dec. (Ours)** | **Lite Enc. (Ours)** | **25.61** | **0.7431** | **0.2474** | **0.1911** | 5.200 | 31.4 | 481.6 | 868.3 |
| | Lite Enc. w/o ASC | 25.42 | 0.7426 | 0.2737 | 0.2050 | 5.329 | 30.7 | 480.8 | 868.3 |
| | Lite Enc. w/o DFI | 25.35 | 0.7427 | 0.2580 | 0.1940 | 5.136 | 31.1 | 478.6 | **867.6** |
| | PixelUnshuffle | 25.23 | 0.7417 | 0.2697 | 0.2028 | 5.552 | **30.5** | **474.8** | 867.7 |
| | SD Enc. | 25.19 | 0.7259 | 0.2685 | 0.1981 | **5.092** | 51.7 | 1032.2 | 901.4 |
| SD Dec. | **Lite Enc. (Ours)** | 25.95 | 0.7503 | **0.2390** | **0.1843** | 6.899 | **58.1** | **1669.7** | **916.2** |
| | SD Enc. | **26.00** | **0.7548** | 0.2445 | 0.1892 | **6.898** | 79.7 | 2221.2 | 949.6 |

loss. To address this, we introduce the Adaptive Skip Connection (ASC), which flexibly transfers essential features to the decoder during encoding. This not only eases the training of the diffusion U-Net but also improves both fidelity and perceptual quality in the super-resolution results.

Additionally, we design the Dual-path Feature Injection (DFI) module to alleviate the information bottleneck between the encoder and U-Net (see Sect. 3.1). In our Dual-path Feature Injection (DFI) design, the lite path provides compressed, information-dense features that offer global structural guidance and align well with the VAE feature distribution in SD, making them easier for the pretrained U-Net to utilize. In contrast, the heavy path contains richer details but lower information density, making it harder for generative models to use directly.

As shown in Table 3, both ASC and DFI consistently improve performance across nearly all metrics with minimal overhead. Figure 5 further proves: removing either module causes visible degradation, particularly in the word "Sausage" (first row) and the plastic chair edges (second row).

**Comparison between the Lite Encoder and Alternative Encoding Strategies.** We compare our lite encoder with the PixelUnshuffle approach and the original SD encoder (SD Enc.). PixelUnshuffle, introduced by AdcSR [23], is a lightweight method for injecting low-resolution (LR) features. Despite similar computational cost, our encoder consistently outperforms PixelUnshuffle across all metrics, with up to **8.3%** improvement in LPIPS. Moreover, our design achieves superior performance on

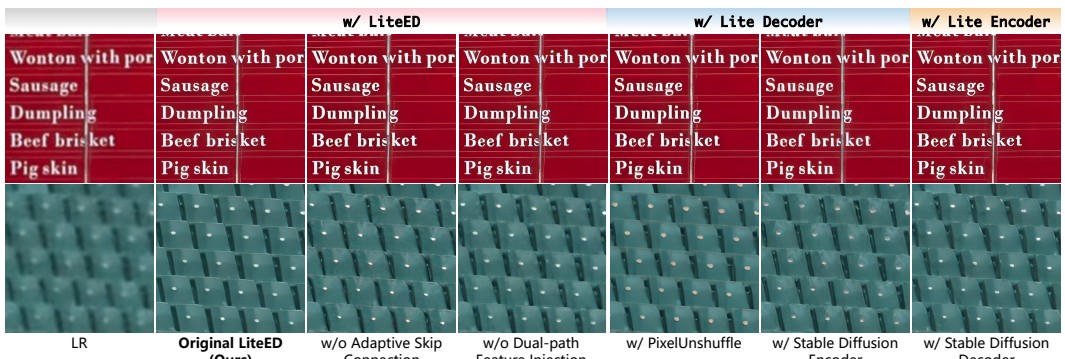

Figure 5: Ablation study of the LiteED design on "Canon 003" from RealSR (top) and "DSC 1286" from DRealSR (bottom).

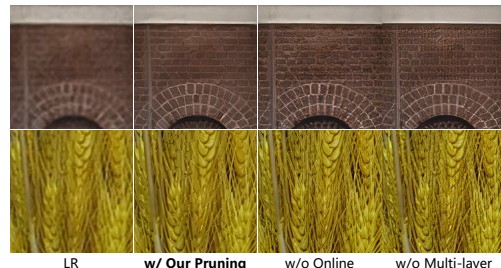

LR | w/ Our Pruning Strategy | w/o Online Annealing Pruning | w/o Multi-layer Feature Distillation

Figure 6: Ablation study of our pruning strategy on "Nikon 004" and "Nikon 015" images from RealSR.

Table 4: Ablation study on pruning strategies. We conduct ablation studies on the online annealing strategy and the multi-layer feature distillation loss employed during pruning. Additionally, we compare the effectiveness of multi-layer distillation against single-layer distillation applied solely to the output layer of the U-Net.

| Strategy | Online Annealing | Multi-layer Distillation | LPIPS↓ | DISTS↓ | NIQE↓ |
|---|---|---|---|---|---|
| Ours | ✓ | ✓ | **0.2713** | **0.2094** | **5.067** |
| (1) | ✗ | ✓ | 0.2732 | 0.2120 | 5.126 |
| (2) | ✓ | ✗ | 0.2816 | 0.2162 | 5.097 |
| (3) | ✓ | Single-layer | 0.2762 | 0.2116 | 5.077 |

reference-based metrics than the SD encoder, with only **46.7%** of its computational cost. We attribute this to the absence of skip connections and the potential bottleneck in the SD encoder, which may result in the loss of fidelity-critical information. As shown in Figure 5, PixelUnshuffle often causes fine detail loss (first row) and oversmoothing (second row), while the SD encoder tends to produce unnatural textures. These results highlight that a well-designed lightweight encoder can outperform complex counterparts in SR tasks, offering valuable insights for efficient model design.

**Compatibility with more powerful image decoders.** Prior studies [70, 23] have shown that decoder complexity often has a greater impact on performance than encoder complexity. Building on this, we investigate the decoder flexibility of LiteED by replacing its decoder with the original SD decoder, while keeping the lite encoder (w/ ASC & DFI) unchanged. This substitution leads to improved SR quality, albeit with a **247%** increase in computational overhead. These results underscore the generalizability of LiteED, enabling flexible decoder adjustment to balance efficiency and performance. We also compare our design with the original SD VAE (w/o ASC & DFI). The variant using the SD decoder and lite encoder achieves comparable performance with a **24.8%** reduction in computational cost, further validating the effectiveness of the LiteED architecture.

**Effect of Online Annealing Pruning.** In Table 4, we compare the proposed online annealing pruning strategy with direct lightweight module replacement, denoted as Strategy (1). Our approach consistently outperforms the baseline across all metrics, especially the no-reference NIQE score, thanks to its progressive knowledge transfer that better preserves the SD model's generative prior. As shown in Figure 6, while direct pruning fails to recover clear brick patterns and wheat textures, our method yields visually superior and high-quality results.

**Effect of Multi-layer Feature Distillation.** Multi-layer feature distillation is introduced to enhance the stability and robustness of knowledge transfer during pruning. As shown in Table 4, removing it (Strategy 2) degrades performance across all metrics, with LPIPS showing the largest drop (**3.8%**). We also evaluate single-layer distillation at the U-Net output (Strategy 3), which slightly outperforms Strategy 2 but still lags behind the full multi-layer setup. As illustrated in Figure 6, removing multi-layer distillation results in noticeable noise and artifacts due to unstable training dynamics.

# 5 Conclusion and limitation

We present PocketSR, a highly efficient single-step model for real-world image super-resolution. By replacing the heavy VAE in SD with LiteED, PocketSR reduces parameters and latency while preserving fidelity. Our proposed pruning strategy further enhances efficiency by progressively transferring generative priors to lightweight modules, optimizing performance without compromising quality. Experiments show that it achieves significant speedup and performance comparable to state-of-the-art RealSR models, making it practical for broad deployment.

One limitation of PocketSR is that its detail generation capability under severe degradations remains to be improved. Moreover, the current framework has not yet been optimized in conjunction with edge hardware, which we identify as a promising avenue for future research.

**Acknowledgments.** This work was supported by the National Key Research and Development Program of China (No. 2024YFB2808903). Thanks for the computing resources provided by Lei Ke during the rebuttal.

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
