# OpenReview forum: "PocketSR: The Super-Resolution Expert in Your Pocket Mobiles"
_NeurIPS.cc/2025/Conference — NeurIPS 2025 poster_

### Official Review · Reviewer_rEby · 2025-07-01

**Clarity:** 3
**Significance:** 3
**Originality:** 2
**Rating:** 5
**Confidence:** 4

**Summary:**

The paper proposes to adapt an existing few-step image generation diffusion model to 1-step image super-resolution, while simultaneously significantly reducing its parameters. First, the encoder and decoder size are drastically reduced to only 2M parameters, followed by the proposed online annealing pruning of the UNet with multi-layer feature distillation. The approach achieves comparable results with SOTA in terms of quality, while reducing the inference time to half of the previous approach at 16 milliseconds for 128x128->512x512 super-resolution.

**Questions:**

- Could the proposed LiteED be used for generation as well?
- The used VAE encode, decode and UNet is a highly symmetric network. However, the low-quality and high-quality images provide inherently non-symmetric information. Was this taken into account in the design for pruning?
- Which findings can be extended to DiTs?

**Ethical Concerns:**

["NO or VERY MINOR ethics concerns only"]

**Final Justification:**

After reading the rebuttal and other reviews I will retain my original score. The authors clarified my concerns, and I believe the method will serve as a strong baseline and most of the proposed techniques can be further extended to stronger base models in the future.

**Limitations:**

Yes

**Quality:**

3

**Strengths And Weaknesses:**

Strengths:
- Comparable results with SOTA with twice the speed and third of the parameters
- A novel approach of transitioning from a few-step generator to a 1-step super-resolution model, rather than previous approaches that start from a multi-step generator and distill it to 1-step.
- Simplicity and generalizability of the proposed online annealing pruning that stabilizes pruning for blocks and channel width

Weaknesses:
- Contributions from theoretical side are somewhat limited. The proposed online annealing pruning seems to be an extension of [2]. Similarly the multi-layer feature distillation has been used in previous work that utilize distillation [4]
- The paper uses "older" base model with an UNet, whereas existing approaches mostly use DiTs for improved image quality, therefore making the contributions not directly applicable
- The choice of pruning seems to have been done somewhat arbitrarily based on high-level findings, rather than careful analysis of MACs and importance of each block.

Minor:
- Include CLIPIQA metric for fairer comparison
- Missing concurrent work PisaSR [1]. Not necessary to add comparison, but it would benefit the community.
- Similar approaches to online annealing pruning have been proposed for channel width compression in [2, 3] that can be mentioned in related work


[1] L Sun, R Wu, Z Ma, S Liu, Q Yi, L Zhang. Pixel-level and Semantic-level Adjustable Super-resolution: A Dual-LoRA Approach, CVPR 2025.
[2] Julian Knodt, Structural Dropout for Model Width Compression, 2022
[3] AN Gomez, I Zhang, SR Kamalakara, D Madaan, K Swersky, Y GAL, GE Hinton. Learning Sparse Networks Using Targeted Dropout, 2019
[4] YLiu, J Cao, B Li, W Hu, J Ding, L Li. Cross-Architecture Knowledge Distillation, ACCV 2022.

---

> ### Author Rebuttal · Authors · 2025-07-31
>
> We sincerely thank the reviewer for recognizing the value of our work and for providing constructive feedback. We will carefully address each of the identified weaknesses and concerns in the following responses. All discussions and additional experiments presented in this rebuttal will be incorporated into the camera-ready version of the paper.
>
> ---
> > **[W1] Contributions of online annealing pruning and multi-layer feature distillation.**
>
> We sincerely thank the reviewer for the valuable comments. However, we believe there may be some misunderstandings regarding the core contributions of our work.
>
> Our main contribution lies in the design of the LiteED architecture, and the integration of Online Annealing Pruning and Multi-layer Feature Distillation, which together enable effective compression for super-resolution (SR) while preserving the generative prior.
>
> 1. While general pruning strategies (as cited in [1,2]) have been explored for efficient text-to-image (T2I) generation, our Figure 6 and Table 4 show that **these general strategies fail to preserve the pretrained prior when applied to SR**—likely due to fundamental differences in training objectives between T2I and SR. To our knowledge, no prior work proposes a pruning strategy specifically tailored to retain image priors in the SR setting, as we do.
>
> 2. Our Multi-layer Feature Distillation further improves prior preservation by supervising the student with multi-scale features across multiple layers. In contrast, **existing methods such as SnapGen[3] and [4] typically distill only from the output**, which imposes a weak constraint. As shown in Table 4, our method outperforms a variant using output-only distillation (Strategy 3), validating its effectiveness under a lightweight setting.
>
> [1] Learning sparse networks using targeted dropout, Arxiv 2019
>
> [2] Structural Dropout for Model Width Compression, Arxiv 2022
>
> [3] Snapgen: Taming high-resolution text-to-image models for mobile devices with efficient architectures and training, CVPR 2025
>
> [4] Cross-Architecture Knowledge Distillation, ACCV 2022
>
>
> ---
> > **[W2] “Older” base model: U-Net.**
>
> We sincerely thank the reviewer for the valuable suggestion. Our choice not to adopt DiT as the base model is primarily driven by deployment considerations. This work aims to reduce computational overhead and enable on-device SR, especially on mobile NPUs, which currently lack sufficient support for DiT-based architectures—making them less practical for edge deployment.
>
> In addition, while some recent single-step SR models use DiT, most state-of-the-art methods—including OSEDiff, AdcSR, and PiSA-SR (as cited by the reviewer)—still rely on U-Net architectures. By using the same backbone, we ensure fair and meaningful comparisons with these widely accepted baselines.
>
>
> ---
> > **[W3] Arbitrary choice of pruning.**
>
> We appreciate the reviewer’s thoughtful analysis. While we understand the concern, we would like to clarify the rationale behind our pruning strategy.
>
> Our pruning design (i.e., where to prune) is guided by empirical observations: shallow modules have greater impact on SR quality, while deeper ones can be pruned with minimal degradation. We support this with ablation studies across different module types and depths, as shown in Tables 1–4 of the supplementary material.
>
> We acknowledge that we did not conduct independent ablations for every single module, mainly for two reasons:
>
> 1. Such exhaustive experiments would require significant resources beyond the scope of this work.
>
> 2. More importantly, we aim to offer conceptual insights rather than purely engineering-driven optimizations, which we believe are more meaningful for the research community.
>
> ---
> > **[W4] CLIPIQA metric.**
>
> We sincerely thank the reviewer for the professional and constructive feedback. We will include a comparison on CLIPIQA in the camera-ready version.
>
> The table below reports the results on the DRealSR dataset. CLIPIQA leverages the pretrained image-text alignment model CLIP to evaluate the consistency between an image and a given textual prompt. **This metric tends to favor results with richer texture details, regardless of their alignment with the ground truth.**
>
> While our method performs slightly worse than OSEDiff and AdcSR on this metric—both of which emphasize stronger generative capability—**we achieve better results on perceptual fidelity metrics such as LPIPS and DISTS, which are more indicative of human-perceived visual quality.** Importantly, all these improvements are achieved under significantly lower computational budgets, highlighting the efficiency and practicality of our approach.
>
> $$
> \begin{array}{l|ccccc}
> \hline
> \textbf{Metrics} & \textbf{FeMaSR} & \textbf{ResShift} & \textbf{OSEDiff} & \textbf{AdcSR} & \textbf{PocketSR} \\\\
> \hline
> \text{CLIPIQA↑} & 0.5464 & 0.5342 & 0.6963 & 0.7049 & 0.6050 \\\\
> \text{LPIPS↓}  & 0.3157 & 0.4006 & 0.2968 & 0.3046 & 0.2962 \\\\
> \text{DISTS↓}  & 0.2239 & 0.2656 & 0.2165 & 0.2200 & 0.2139 \\\\
> \text{FPS↑}    & 16.8   & 1.4    & 9.1    & 33.3   & 62.5   \\\\
> \hline
> \end{array}
> $$
>
> ---
> > **[W5&W6] Missing citations.**
>
> We sincerely thank the reviewer for the helpful suggestion. We will cite the papers mentioned by the reviewer in the Related Work section of the camera-ready version.
>
> ---
> > **[Q1] Could the proposed LiteED be used for generation?**
>
> We thank the reviewer for the thoughtful and forward-looking question.
>
> **The proposed lite decoder can be directly applied to efficient image generation tasks.** However, the lite encoder includes components specifically tailored for super-resolution—such as Adaptive Skip Connections and Dual-Path Feature Injection—which may not transfer well to generative settings and would likely require adaptation.
>
> In image generation, encoders and decoders are typically used asymmetrically: the encoder is only used during training, while the decoder is involved in both training and inference. This prevents the use of skip connections as employed in super-resolution.
>
> Additionally, generative models often operate in compact latent spaces. The heavy-path features from our encoder may introduce redundancy that could hinder generation quality rather than help it.
>
> ---
> > **[Q2] Network symmetry.**
>
> Thank you for the excellent question. We explicitly considered this asymmetry when designing LiteED. Empirically, we observed that in super-resolution, the decoder plays a more critical role than the encoder in determining performance.
>
> As shown in Table 1, the decoder accounts for 8.7× more MACs than the encoder. Moreover, Table 3 shows that enlarging the decoder significantly improves reconstruction quality, while increasing encoder complexity yields only marginal gains and adds training overhead.
>
> This likely stems from the inherent nature of SR: low-resolution inputs contain less information than the high-resolution outputs. **Under a fixed computational budget, it is thus more effective to prioritize the decoder, which reconstructs the richer content.**
>
>
> ---
> > **[Q3] Can findings be extented to DiTs?**
>
> Thank you for the insightful question. We agree that extending our approach to a DiT-based backbone for lightweight image super-resolution is a promising direction. **In principle, both components of our method—LiteED and online annealing pruning—are transferable to other base models.** LiteED targets the VAE-style decoder, and the pruning strategy is architecture-agnostic.
>
> However, certain pruning decisions—such as where to prune and which lightweight modules to use—would need to be revisited for DiT, taking its structural characteristics into account. Recent work [5] suggests that DiT exhibits layer-wise specialization, with intermediate layers contributing most to semantic alignment. Such insights could help guide pruning when adapting our method to DiT.
>
> We appreciate the reviewer’s suggestion, which points to a valuable future direction for improving the generalizability of our framework.
>
> [5] Revelio: interpreting and leveraging semantic information in diffusion models, Arxiv 2024

---

> > ### Author Response · Authors · 2025-08-05
> > **Have our responses successfully addressed your concerns?**
> >
> > We sincerely appreciate the time and effort you dedicated to reviewing our paper. Several of your comments were particularly insightful and thought-provoking. We have carefully responded to each of the weaknesses and suggestions you raised. If you have a moment, could you kindly let us know whether our responses have successfully addressed your concerns? Your feedback is very important to us. Thank you again for your thoughtful contributions.

---

> > > ### Comment · Reviewer_rEby · 2025-08-05
> > >
> > > Thank you for the clarifications.
> > >
> > > - For W3 I do not think exhaustive architecture search with pruning would be necessary, but for example deactivating individual blocks during inference and analyzing its performance degradation. Although not necessarily as extensive as pruning, it could give stronger guidance compared to the existing approach. I do however agree on the importance of conceptual insights provided currently.
> > >
> > > - Other reviewers made comments regarding the applicability to mobile, which is of course important. Due to the complexity of implemention on mobile I believe the current results are sufficient. The reported time was 140ms, whereas AdcSR achieves 67ms for the same task (Table C.3 [AdcSR]) on a Snapdragon 8 Gen 4 using W8A16 requiring 500MB of memory.
> > >    * Could you also share the memory requirements?
> > >    * What could be reason for the discrepancy? NPU efficiency? Lack of efficient attention implementation?
> > >
> > > - I also appreciate the authors willingness to open source the code despite limitations.
> > >
> > > Thank you.

---

> > > > ### Author Response · Authors · 2025-08-06
> > > > **Response to Reviewer rEby**
> > > >
> > > > Thank you for your professional and insightful feedback. We agree that conducting a finer-grained search, when resources permit, could further improve model performance, as different modules at the same depth may contribute unequally. We consider this an important direction for future work.
> > > >
> > > > Our model contains 146M parameters. During on-device inference, additional memory is required to cache certain skip features, resulting in a total memory footprint of approximately 620MB.
> > > >
> > > > Regarding inference time on edge devices, we acknowledge the reviewer’s observation. As noted, our evaluation was conducted on a self-developed NPU, which is less powerful than the Snapdragon 8 Gen 4, leading to longer measured runtimes. Additionally, on-device latency is influenced by various factors such as NPU model, operating temperature, and chip frequency. Without a standardized testing environment, cross-device latency comparisons can be difficult to interpret fairly.
> > > >
> > > > We warmly welcome further evaluation on Snapdragon 8 Gen 4 once our implementation is open-sourced.

---

> > > > > ### Comment · Reviewer_rEby · 2025-08-06
> > > > >
> > > > > Thank you for the clarifications.

---

### Official Review · Reviewer_nGC4 · 2025-07-02

**Clarity:** 2
**Significance:** 4
**Originality:** 2
**Rating:** 4
**Confidence:** 4

**Summary:**

This work proposes PocketSR, a lightweight, single-step, diffusion-based model for image super-resolution.
This paper introduces three key contributions:

(i) LiteED, which is an efficient encoder-decoder architecture that significantly reduces the parameter count of the standard SD VAE by 97.5%. It incorporates dual-path feature injection to enrich the U-Net with high-dimensional features and adaptive skip connections to retain critical information, mitigating the impact of the compressed encoder.

(ii) Online annealing pruning for U-Net, which progressively shifts generative priors from heavier, original U-Net modules to lightweight replacements. This method ensures a stable transfer of knowledge, preserving the generative capability. The paper also identifies that pruning deeper layers of the U-Net has a minimal impact on performance, offering an effective strategy for model compression.

(iii) Multi-layer Feature Distillation: This technique is used in conjunction with online annealing pruning to facilitate a more stable and reliable knowledge transfer from the larger model to the pruned one, preserving the generative priors and avoids the instability that can occur with single-layer distillation.

**Questions:**

1. As mentioned in weakness, please consider report mobile benchmark.
2. What is the detailed structure for the "lightweight" block? Is it the same for VAE and Unet? Is it Mobilenet-like? What are the norm layer and activations? And what is the ultra-lightweight FFN in line 158?
3. In line 159 what is the linear attention employed here? There are different variants. And how about the tradeoff here? Is it difficult to adapt?

**Ethical Concerns:**

["NO or VERY MINOR ethics concerns only"]

**Final Justification:**

The authors addressed most of my concerns in the rebuttal, after reading other reviews, I will keep my voting for acceptance.

**Limitations:**

Yes

**Quality:**

4

**Strengths And Weaknesses:**

Strength:
1. The motivation and proposed methods are technically sound and easy to follow.
2. Efficient diffusion-based SR is an important application, the investigation of its efficiency and quality is timely and of great research value.
3. The empirical results are comprehensive. Under significant efficiency gain, the SR quality still achieves state-of-the-art.
4. The authors provide comprehensive ablation studies that validate each of their design choices including adaptive skip connection, dual-path feature injection, and the online annealing pruning strategy.

Weakness:
1. The authors state that this work will not be open-sourced, which limits the reproducibility. In fact as a low-level vision application, opensourcing the model can benefit the community a lot, while having less risks compared to generative models.
2. There is a mismatch between the claimed "pocket" scope and the reported server GPU benchmarks, which is problematic. I think this work should either report mobile benchmark, or re-scope the contribution to "GPU-efficient" SR.
3. Novelty is a minor concern. The methods in this work (pruning, feature distillation, step distillation) are not brand new. Overall the successful integration of these methods to obtain an efficient, single-step diffusion-based SR model can be viewed as sufficient contribution, but not quite novel.

---

> ### Author Rebuttal · Authors · 2025-07-31
>
> We sincerely thank the reviewer for the time and effort dedicated to evaluating our work. We are encouraged to see that our paper was described as “technically sound” and of “great research value.” We will address each of the reviewer’s concerns and identified weaknesses in detail. All additional experiments and discussions presented in this rebuttal will be incorporated into the camera-ready version of the paper.
>
> ---
> > **[W1] Reproducibility.**
>
> We fully understand and agree with the reviewer’s concern. In response, we have decided to conduct a code reimplementation, within the boundaries of our company’s policy, and plan to release it publicly upon acceptance to better support the community.
>
> ---
> > **[W2&Q1] Efficiency on mobile devices.**
>
> We fully understand the reviewer’s concern. However, we hope you can appreciate that migrating and deploying all existing super-resolution methods to mobile devices would require a significant amount of time and engineering effort. As an alternative, we provide the measured inference efficiency of PocketSR on a newly released smartphone model from 2025. For comparison, we also report the runtime of the uncompressed SD-Turbo backbone on the same device across corresponding modules.
>
> The results are shown in the table below (measured on a single 512×512 image; unit: milliseconds). As can be observed, **PocketSR achieves an inference time of only 140ms on-device, representing an over 80% reduction compared to the original, non-lightweight backbone**—demonstrating the practical efficiency of PocketSR for real-world deployment.
>
> $$
> \begin{array}{l|rrrr}
> \hline
> \textbf{Models} & \textbf{Encoder} & \textbf{U\-Net} & \textbf{Decoder} & \textbf{Total} \\\\
> \hline
> \text{SD-Turbo} & 320ms & 270ms & 200ms & 790ms \\\\
> \text{PocketSR} & \textbf{\downarrow 97 \\%} \\:11ms & \textbf{\downarrow 67 \\%}\\:88ms & \textbf{\downarrow 80 \\%}\\:41ms & \mathbf{\downarrow 82\\%}\\:140ms \\\\
> \hline
> \end{array}
> $$
>
> ---
> > **[W3] Concern on novelty.**
>
> We sincerely thank the reviewer for the thoughtful question. However, we believe there may have been a misunderstanding regarding the core contributions of our work.
>
> Our paper makes three primary contributions:
>
> 1. We propose **a lightweight encoder-decoder framework, LiteED, specifically designed for the super-resolution task**. LiteED integrates carefully designed components, including Adaptive Skip Connections and Dual-Path Feature Injection, and serves as an effective replacement for the original SD VAE. Remarkably, despite its significantly reduced parameter count, LiteED matches or even outperforms the original VAE in certain aspects.
> 2. We introduce **Online Annealing Pruning, a pruning strategy tailored for low-level vision tasks**. While similar pruning concepts have been explored in efficient text-to-image (T2I) generation, our Figure 6 and Table 4 show that such methods fail to retain the image priors encoded in pretrained generative models when applied to super-resolution. This discrepancy stems from the fundamental difference between T2I and SR training objectives.
> 3. Our proposed **Multi-layer Feature Distillation enhances the transfer of generative knowledge by guiding the student model with multi-scale features across several layers**. In contrast, prior methods typically distill only from the model’s output—a relatively weak constraint that may fail to preserve rich prior information after pruning. In Table 4, we compare our approach with a variant that distills solely from the U-Net output (Strategy 3). The results clearly demonstrate the superior performance of our method, validating its effectiveness in retaining the pretrained generative prior in a lightweight setting.
>
> To the best of our knowledge, all contributions are novel and have not been addressed in prior work. While we do adopt step distillation, it is not the core innovation of our paper.
>
> ---
> > **[Q2] Detailed structure for the “lightweight” block.**
>
> Thank you for the insightful question. We provide an explanation of the “Lightweight Block” in Figure 2 (lines 153–160), and would like to clarify the following:
>
> 1. **Scope of Pruning:** The Online Annealing Pruning strategy is applied only to the U-Net, not to LiteED, so the notion of “Lightweight Block” does not apply to LiteED.
>
> 2. **Pruned Components:** Pruning targets the four most computationally expensive U-Net components: residual blocks, cross-attention, self-attention, and FFNs. Lightweight layers (e.g., normalization, activation) remain unchanged.
>
> 3. **Pruning Details:**
>
>    * **Residual Blocks:** All convolutions are replaced with depthwise separable convolutions (as in MobileNet), forming the “Lightweight Block.”
>    * **Cross-Attention:** Since text input is not used in SR, we replace these layers with an MLP of two linear layers.
>    * **Self-Attention:** Approximated using linear attention to reduce cost.
>    * **FFNs:** Hidden dimensions are reduced to one-fourth.
>
> We acknowledge that the term “ultra-lightweight FFN” may have caused confusion and will revise it for clarity in the camera-ready version.
>
> ---
> > **[Q3] Questions on linear attention.**
>
> Thank you for the question. We adopt the basic form of linear attention as described in [1].
>
> As shown in Table 3 of the supplementary material, replacing standard attention with linear attention involves performance trade-offs. Our experiments indicate that self-attention—especially in the shallow layers of the U-Net—is crucial for capturing fine-grained textures. Replacing it with linear attention in early layers leads to noticeable degradation, likely because these layers extract low-level features that lightweight alternatives cannot easily replicate.
>
> To balance efficiency and performance, we apply linear attention only at depth 4 of the U-Net. This modification preserves model performance while improving training stability and efficiency.
>
> [1] Transformers are RNNs: Fast Autoregressive Transformers with Linear Attention, ICML 2020

---

> ### Author Response · Authors · 2025-08-05
> **Have our responses successfully addressed your concerns?**
>
> We sincerely appreciate the time and effort you dedicated to reviewing our paper. Several of your comments were particularly insightful and thought-provoking. We have carefully responded to each of the weaknesses and suggestions you raised. If you have a moment, could you kindly let us know whether our responses have successfully addressed your concerns? Your feedback is very important to us. Thank you again for your thoughtful contributions.

---

> > ### Comment · Reviewer_nGC4 · 2025-08-07
> >
> > Thanks for providing the rebuttal. After reading the author's clarifications and other reviews, most of my concerns have been addressed. I appreciate the authors including the actual mobile benchmark, and encourage the authors to include a bit more comparisons in the revision especially comparisons to non-diffusion SR models, so that readers can get a better understanding of the performance and speed tradeoff.
> >
> > One followup comment on Q3, if linear attention is only applied in depth 4, then the speed gain is minimal. It would be better to include more discussions in the revision about this design choice and tradeoff.
> >
> > I tend to keep my positive rating.

---

> > > ### Author Response · Authors · 2025-08-07
> > > **Response to Reviewer nGC4**
> > >
> > > Thank you for your constructive feedback. We’re glad to hear that our response helped clarify your concerns. We will include the mobile benchmark results and the comparison with non-diffusion-based methods in the revised version of the paper.
> > >
> > > Regarding your observation about applying linear attention only at depth 4: we found that replacing it with linear attention in early layers leads to noticeable degradation, likely because these layers are responsible for capturing low-level features that lightweight alternatives struggle to replicate. As a result, the efficiency gain from linear attention is smaller compared to other components. We will add a detailed explanation of this observation to the revised paper. However, it's important to clarify that our optimization is not limited to self-attention modules. From an overall perspective, the efficiency gain brought by our online annealing pruning strategy remains significant.
> > >
> > > We sincerely appreciate your feedback. Please feel free to reach out if you have any further questions or thoughts.

---

### Official Review · Reviewer_2VLv · 2025-07-02

**Clarity:** 2
**Significance:** 3
**Originality:** 2
**Rating:** 5
**Confidence:** 5

**Summary:**

PocketSR is a lightweight and efficient model for the RealSR task, using a compact encoder-decoder and efficient pruning to reduce size and latency. It introduces LiteED, a compact alternative to the VAE used in Stable Diffusion, reducing parameters by 97.5% while maintaining high quality. The model also uses online annealing pruning and multi-layer feature distillation to optimize performance and retain generative knowledge. With only 146M parameters, PocketSR can process 4K images in 0.8 seconds, offering speed and quality comparable to state-of-the-art methods, making it ideal for real-time mobile and edge applications.

**Questions:**

1. How to obtain the evaluation values in Table 3? Are they used to evaluate the reconstruction results of the VAE? What test dataset was used? For the encoders and decoders compared with LiteED, were they all trained together with the unpruned SD U-Net for 80000 steps as well?

2. Which version of Stable Diffusion is PocketSR based on?

3. The paper repeatedly emphasizes PocketSR's speed advantage in processing 4K images. What is the performance of PocketSR in handling 4K images SR? Is its performance on different resolutions affected by the choice of the pretrained SD model?

4. The design of ASC is interesting, but it requires more detailed ablation studies and explanations to give readers a clearer understanding:

A. Compare the performance between skip connections (without the control coefficients) and adaptive skip connections.

B. Why are the control coefficients learned from the output of the lite path? What would happen if they were learned from the heavy path instead?

C. What are the characteristics of the learned control coefficients? Do they show any patterns based on different input contents or statistical properties?

D. During inference, how would modifying the control coefficients affect the output?

5. The purpose of the dual-path design is not clearly explained. What are the respective roles of the lite path and the heavy path? What would happen to performance if either the lite or heavy path were removed?

**Ethical Concerns:**

["NO or VERY MINOR ethics concerns only"]

**Final Justification:**

Most of my concerns have been satisfactorily addressed. I encourage the authors to provide further clarification on the DFI and ASC components in the revised version to enhance clarity. Additionally, applying the proposed strategy to a more recent model, such as FLUX, could further strengthen the contribution.

After this discussion, I have decided to raise my rating from Borderline Accept to Accept.

**Limitations:**

Yes.

**Paper Formatting Concerns:**

No.

**Quality:**

3

**Strengths And Weaknesses:**

Strengths:

1. PocketSR demonstrates impressive speed, making it suitable for real-time applications on edge devices.

2. The introduction of LiteED significantly reduces model size and computational cost while maintaining high-quality results.

3. The paper proposes novel methods, such as online annealing pruning and adaptive skip connections (ASC), which contribute to both efficiency and performance.

4. Despite its simplicity and single-step design, PocketSR performs on par with more complex, multi-step RealSR models.

Weaknesses:

1. The methodology behind some evaluations (e.g., Table 3) is unclear, specifically which datasets were used, and whether all encoder-decoder pairs were trained under the same conditions.

2. The paper doesn't specify which version of Stable Diffusion PocketSR is based on.

3. While speed is emphasized, the quality of 4K image super-resolution results is not thoroughly analyzed. It’s also unclear how performance varies across resolutions or with different pretrained SD models.

4. The ASC module is underexplored. Key aspects such as the function of control coefficients and their effect during inference are not fully examined.

5. The roles of the lite path and heavy path in the dual-path structure are not well justified. The impact of removing either path is not studied.

---

> ### Author Rebuttal · Authors · 2025-07-31
>
> We sincerely thank the reviewer for the insightful and constructive feedback, which has prompted us to further reflect on our approach. We also appreciate your recognition of our work’s effectiveness and novelty. Below, we address your concerns point by point (with related issues grouped for clarity). All discussions and new experiments will be included in the camera-ready version.
>
> ---
> > **[W1&Q1] Explanation for Table 3.**
>
> We apologize for not clearly specifying the experimental setup of Table 3, and we will include this information in the camera-ready version. The experiments in Table 3 follow the first-stage training described in Section 3.3, where the unpruned SD U-Net is jointly trained with various encoder and decoder architectures for 80,000 steps.
>
> Table 3 evaluates the impact of different encoder and decoder designs on super-resolution performance (not VAE reconstruction), serving as an ablation for our proposed LiteED. U-Net pruning is not applied at this stage. All settings share the same model and training configuration, differing only in encoder/decoder architecture. RealSR is used as the test dataset.
>
> ---
> > **[W2&Q2] Explanation for SD version.**
>
> Thank you for the question. As mentioned in line 101 of the paper, PocketSR uses SD-Turbo as the pretrained model.
>
> ---
> > **[W3&Q3] Performance on handling high-resolution SR.**
>
> Thank you for the valuable suggestion. To assess PocketSR at higher resolutions, we constructed a multi-resolution benchmark with three 4× SR tasks: upscaling to 1K, 2K, and 4K. Each setting includes 20 test images with ground-truths. The 1K and 2K samples are selected from RealSR, ensuring resolution and scene diversity; 4K samples are from DRealSR. Notably, this benchmark uses original full-resolution images, unlike Table 2 which evaluates on center-cropped patches—a common practice in SR [1].
>
> As shown below, PocketSR consistently achieves superior LPIPS, DISTS, and NIQE scores, and competitive PSNR, SSIM, and MUSIQ results. Its stable performance across resolutions demonstrates strong robustness, making it well-suited for real-world use cases like mobile photography.
>
> $$
> \begin{array}{l|l|cccccc}
> \hline
> \textbf{Tasks} & \textbf{Methods} & \textbf{PSNR} \uparrow & \textbf{SSIM} \uparrow & \textbf{LPIPS} \downarrow & \textbf{DISTS} \downarrow & \textbf{NIQE} \downarrow & \textbf{MUSIQ} \uparrow \\\\
> \hline
> \text{1K Resolution SR} & \text{SinSR} & \mathbf{25.53} & 0.7015 & 0.3934 & 0.2286 & 5.794 & 62.82 \\\\
> & \text{OSEDiff} & 24.73 & 0.7088 & 0.3226 & 0.1933 & 3.943 & 70.50 \\\\
> & \text{AdcSR} & 25.01 & 0.6995 & 0.3154 & 0.1886 & \mathbf{3.409} & \mathbf{72.05} \\\\
> & \text{PocketSR (Ours)} & 25.07 & \mathbf{0.7162} & \mathbf{0.2709} & \mathbf{0.1662} & 3.880 & 65.23 \\\\
> \hline
> \text{2K Resolution SR} & \text{SinSR} & \mathbf{26.96} & 0.7473 & 0.3884 & 0.2102 & 6.563 & 55.25 \\\\
> & \text{OSEDiff} & 25.81 & \mathbf{0.7756} & 0.2775 & 0.1661 & 4.333 & 63.78 \\\\
> & \text{AdcSR} & 26.35 & 0.7732 & 0.2729 & 0.1630 & 4.158 & 63.52 \\\\
> & \text{PocketSR (Ours)} & 26.41 & 0.7736 & \mathbf{0.2467} & \mathbf{0.1510} & \mathbf{4.060} & \mathbf{63.85} \\\\
> \hline
> \text{4K Resolution SR} & \text{SinSR} & 27.36 & 0.7086 & 0.4753 & 0.2379 & 6.246 & 29.56 \\\\
> & \text{OSEDiff} & 26.37 & \mathbf{0.7951} & 0.3200 & 0.1696 & 4.291 & \mathbf{36.69} \\\\
> & \text{AdcSR} & \mathbf{27.37} & 0.7888 & 0.3258 & 0.1784 & 4.325 & 34.34 \\\\
> & \text{PocketSR (Ours)} & 26.76 & 0.7752 & \mathbf{0.3174} & \mathbf{0.1664} & \mathbf{4.064} & 34.25 \\\\
> \hline
> \end{array}
> $$
>
> We adopt SD-Turbo for SD pretraining, a widely used default in many super-resolution frameworks [2,3]. The reviewer raises an insightful question about whether model performance across resolutions is influenced by the resolution of the pretrained SD. We believe the answer is likely yes, as SD models trained on different resolutions may encode distinct resolution-specific priors. We consider this a valuable direction for future work.
>
> [1] Exploiting Diffusion Prior for Real-World Image Super-Resolution, IJCV 2024
>
> [2] Arbitrary-steps image super-resolution via diffusion inversion, CVPR 2025
>
> [3] Degradation-Guided One-Step Image Super-Resolution with Diffusion Priors, Arxiv 2024
>
> ---
> > **[W4&Q4-1] Detailed ablation studies for the Adaptive Skip Connection (ASC).**
>
> We sincerely thank the reviewer for the constructive suggestions. In response, we provide additional ablation studies on the Adaptive Skip Connection (ASC) module in the table below. The experiments are conducted on the RealSR dataset, following the same settings as in Table 3 of the main paper for ease of comparison. We omit efficiency-related metrics here, as the differences are minimal.
> 1. We investigate the effect of removing the learnable control coefficients in ASC. We observe that using naive skip connections hinders the model’s ability to adaptively control the contribution of skip features based on the degradation level of the input. As a result, the model performs poorly when handling diverse real-world degradations.
> 2. We explore an alternative design where the control coefficients are predicted from features in the heavy path. However, we find that this strategy leads to instability during training, as heavy path features are uncompressed and contain redundant information, which interferes with the learning of accurate control signals. Although this variant performs relatively well on the no-reference metric NIQE, **the significantly lower LPIPS and PSNR scores suggest compromised perceptual and pixel-level fidelity**, with more noticeable artifacts in the output.
>
> $$
> \begin{array}{l|ccccc}
> \hline
> \textbf{LiteED Design} & \textbf{PSNR} \uparrow & \textbf{SSIM} \uparrow & \textbf{LPIPS} \downarrow & \textbf{DISTS} \downarrow & \textbf{NIQE} \downarrow \\\\
> \hline
> \text{Original} & \mathbf{25.61} & \mathbf{0.7431} & \mathbf{0.2474} & \mathbf{0.1911} & 5.200 \\\\
> \text{w/o control coefficients in ASC} & 25.16 & 0.7276 & 0.2486 & 0.1999 & 5.321 \\\\
> \text{w/ control coefficients learning from the heavy path} & 25.08 & 0.7347 & 0.2667 & 0.1995 & \mathbf{4.952} \\\\
> \hline
> \end{array}
> $$
>
> ---
> > **[W4&Q4-2] Characteristics of the learned control coefficients.**
>
> Thank you for the excellent question. Due to space constraints, we couldn’t include this analysis in the paper, but we provide more details below:
>
> 1. Control Coefficients vs. Outputs: Larger control coefficients make outputs more similar to the low-resolution (LR) input—preserving fidelity but reducing generative strength. Smaller coefficients shift reliance to the U-Net output, enhancing generative capacity at the cost of fidelity.
> 2. Control Coefficients vs. Inputs: For severely degraded LR inputs (e.g., high-scale SR, real-world degradations), control coefficients are generally smaller in early layers and slightly larger in deeper ones. In mild degradation cases, the pattern reverses, with higher coefficients in shallow layers.
>
> This behavior aligns with intuition: mildly degraded inputs benefit from more skip features to preserve details, while heavily degraded ones require suppressing skips to rely more on generative priors for plausible reconstruction.
>
> Regarding Q4-D, we agree that manually adjusting the control coefficients can reveal a similar fidelity–generation trade-off. However, such tuning is sensitive and may cause instability, so we recommend using the model-learned adaptive coefficients, which are more robust across varied degradations.
>
> ---
> > **[W5&Q5] Explanation and ablation studies for the Dual-path Feature Injection (DFI).**
>
> Thank you for the thoughtful question. In our Dual-path Feature Injection (DFI) design, the lite path provides compressed, information-dense features that offer global structural guidance and align well with the VAE feature distribution in SD, making them easier for the pretrained U-Net to utilize. In contrast, the heavy path contains richer details but lower information density, making it harder for generative models to use directly.
>
> **The lite path serves as the primary guidance source, while the heavy path complements it with fine-grained textures. Both are essential for high-quality super-resolution.** To validate this, we conducted ablation studies following the experimental settings in Table 3 of the main paper. We observe that removing the lite path leads to a significant drop in fidelity, as the model loses its primary information source. In this case, the model overly relies on the pretrained SD’s generative prior, often hallucinating textures, which results in perceptual artifacts and inflated non-reference scores like NIQE. On the other hand, removing the heavy path causes a moderate degradation in detail reconstruction, reflected by a noticeable drop in perceptual metrics such as LPIPS and DISTS.
>
> $$
> \begin{array}{l|ccccc|ccc}
> \hline
> \textbf{LiteED Design} & \textbf{PSNR} \uparrow & \textbf{SSIM} \uparrow & \textbf{LPIPS} \downarrow & \textbf{DISTS} \downarrow & \textbf{NIQE} \downarrow & \textbf{Time (ms)} & \textbf{MACs (G)} & \textbf{Param. (M)} \\\\
> \hline
> \text{Original DFI} & \mathbf{25.61} & \mathbf{0.7431} & \mathbf{0.2474} & \mathbf{0.1911} & 5.200 & 31.4 & 481.6 & 868.3 \\\\
> \text{w/o Lite Path} & 24.75 & 0.7229 & 0.2596 & 0.1991 & \mathbf{4.860} & 31.3 & 481.6 & 868.3 \\\\
> \text{w/o Heavy Path} & 25.35 & 0.7427 & 0.2580 & 0.1940 & 5.136 & \mathbf{31.1} & \mathbf{478.6} & \mathbf{867.6} \\\\
> \hline
> \end{array}
> $$
>
>
> In summary, DFI effectively balances structural guidance and detail injection, significantly improving fidelity with minimal computational overhead.

---

> > ### Author Response · Authors · 2025-08-05
> > **Have our responses successfully addressed your concerns?**
> >
> > We sincerely appreciate the time and effort you dedicated to reviewing our paper. Several of your comments were particularly insightful and thought-provoking. We have carefully responded to each of the weaknesses and suggestions you raised. If you have a moment, could you kindly let us know whether our responses have successfully addressed your concerns? Your feedback is very important to us. Thank you again for your thoughtful contributions.

---

> > ### Comment · Reviewer_2VLv · 2025-08-06
> >
> > Thank you to the authors for the detailed response. The rebuttal addresses many of the key concerns. I hope the authors can include the following in the revised version: a thorough explanation of the experimental settings for Table 3, as well as a clearer description of the respective roles of the heavy path and lite path.
> >
> > I also have two follow-up questions:
> > 1. During the training of LiteED, are both the lite path and the heavy path used? What is the size of the input to the Decoder?
> >
> > 2. Why does the lite path align well with the VAE feature distribution? Is this related to the way LiteED is trained?

---

> > > ### Author Response · Authors · 2025-08-06
> > > **Response to Reviewer 2VLv**
> > >
> > > Thank you for your feedback. We are glad that our response helped clarify your concerns. We will incorporate the detailed experimental settings for Table 3, along with a clearer explanation of the roles of the heavy path and lite path, into the revised version of our paper.
> > >
> > > During the training of LiteED, both the heavy and lite paths are utilized to ensure that the model fully leverages both types of features for reconstruction. The input resolution during training is 512×512, and at this resolution, the input to the lite decoder is of size B×4×64×64 (where B denotes the batch size).
> > >
> > > Regarding the features from the lite path, we did not explicitly align them with those from the original SD VAE encoder. This decision is based on two main considerations:
> > > 1. Our model includes several adaptive skip connections that do not exist in the original SD VAE. These connections influence the extracted features—functioning similarly to residual paths—and may disrupt direct alignment.
> > > 2. The lite encoder has a very small parameter count, making it difficult to achieve good alignment performance through direct feature matching.
> > >
> > > In our previous response, when we mentioned that the lite path "aligns well with the VAE feature distribution," our intention was to convey that **both the lite path and the original SD VAE encoder produce compressed, information-dense features**. Compared to the heavy path features—which are lower-level and richer in detail—these compressed features are more readily utilized by the pretrained U-Net.
> > >
> > > P.S. Although we do not perform explicit alignment at the encoder output, we do align the decoder input features with those from the original SD VAE (ground truth), in order to prevent the model from drifting away from the original SD feature space during training.

---

> ### Comment · Reviewer_2VLv · 2025-08-06
>
> Thank you to the authors for the response. Most of my concerns are solved.
>
> I still have a few questions for further discussion.
>
> 1. The output of the U-Net has a size of B×4×64×64, which matches the size of the output from the lite path in the encoder. Do the authors observe that this design may lead to some additional information introduced by the heavy path—through the increased channel width of 512—being ultimately discarded?
>
> 2. In the ablation study, the model without the lite path performs worse than the one without the heavy path. Could the authors elaborate on the underlying reasons for this result?  Does this indicate that the lite path plays a more critical role than the heavy path, perhaps because its output directly matches the input dimensions of the decoder? In the authors' response, the lite path is described as the primary source of information—could the authors clarify why this is the case?
>
> Additionally, in pre-trained diffusion models, the VAE often results in information loss, which can negatively impact restoration tasks such as super-resolution. What are the authors' thoughts on the future development direction of diffusion-based super-resolution models?

---

> ### Author Response · Authors · 2025-08-07
> **Response to Reviewer 2VLv**
>
> Thank you for the constructive discussion.
>
> ---
> > **[Q1] The size of the U-Net ouput.**
>
> First, we would like to clarify the motivation behind using 4 channels at the U-Net output. This design choice was made to facilitate alignment with the pretrained decoder weights, making the learning process smoother. As mentioned in our previous response, we align the decoder input features with the latent space of Stable Diffusion (SD), and using a consistent feature dimension helps stabilize supervision during training.
>
> Furthermore, we believe that introducing heavy-path features is meaningful—even though their spatial and channel dimensions are bigger than the U-Net output's. This is based on the observation that SD is capable of generating rich and detailed 512×512 images from 4-channel latent features. **The transformation from heavy-path input to U-Net output only increases feature density, not information loss.** This differs from the encoder side: the lightweight encoder design (only 0.8M parameters) may struggle to extract sufficient detail, and thus the heavy path serves as a necessary supplement.
>
> Of course, we agree with your observation—increasing the number of latent channels can indeed be beneficial, and this aligns with the evolving trends in modern image generation models. For example, the more recent FLUX model uses a latent space with 16 channels. However, since our current design is based on Stable Diffusion (SD), we use 4 channels to remain compatible with the pretrained decoder. In the future, we plan to explore updated backbone models and consider using higher-dimensional latent representations.
>
> ---
> > **[Q2] Explanation of the ablation study.**
>
> Regarding the ablation results where removing the lite path leads to a larger performance drop than removing the heavy path, we believe this is because our model builds upon pretrained T2I weights. After fine-tuning, the model mainly relies on the compressed information flow through the lite path. When the lite path is removed, the model becomes entirely dependent on the uncompressed heavy-path features, which contain redundant information and interfere with the learning of accurate control signals. This observation further validates our dual-path design, which enables complementary feature injection from both directions.
>
> ---
> > **[Q3] Future direction.**
>
> As for the reviewer’s question on VAE-related information loss, we agree that this is a well-known issue in current SR models. On the one hand, recent progress in VAE design—such as the FLUX series—has significantly improved restoration capacity. On the other hand, strategies such as using VAEs with lower compression ratios (e.g., 4× downsampling instead of 8×), or leveraging dual-path feature injection as proposed in our work, may help mitigate this problem.
>
> In terms of future directions, we believe that enhancing the generative capacity of models while achieving greater efficiency will be a key research direction in the future. Specific to the problem, we think semantic information remains a critical component in activating the generative potential of SR models. Existing semantic extractors, such as LLaVA (used in SUPIR), cognitive encoders (CoSeR), and DAPE (SeeSR), are often too heavy for practical use. Developing lightweight yet effective semantic extraction methods to guide texture generation in a perceptually aligned manner remains an important open challenge.

---

> > ### Comment · Reviewer_2VLv · 2025-08-07
> >
> > Thank you to the authors for their detailed response. Most of my concerns have been satisfactorily addressed. I encourage the authors to provide further clarification on the DFI and ASC components in the revised version to enhance clarity. Additionally, applying the proposed strategy to a more recent model, such as FLUX, could further strengthen the contribution.
> >
> > After this discussion, I have decided to raise my rating from Borderline Accept to Accept.

---

> > > ### Author Response · Authors · 2025-08-07
> > > **Response to Reviewer 2VLv**
> > >
> > > Thank you for the ongoing and constructive discussion. Our exchange has also prompted further reflection on our own work. We’re glad to hear that we were able to address your concerns. The clarifications regarding DFI and ASC will be incorporated into the revised version of the paper. If you have any further questions, we would be happy to continue the discussion.

---

### Official Review · Reviewer_2iJ7 · 2025-07-03

**Clarity:** 3
**Significance:** 3
**Originality:** 2
**Rating:** 4
**Confidence:** 5

**Summary:**

This paper introduces PocketSR, an ultra-lightweight diffusion-based model for real-world image super-resolution targeting edge deployment. The core methods include:
LiteED: A highly efficient VAE alternative reducing parameters by 97.5% while preserving encoding/decoding quality via adaptive skip connections and dual-path feature injection.
Online Annealing Pruning: A strategy to progressively shift generative priors from heavy U-Net modules to lightweight counterparts, minimizing knowledge loss.
Multi-layer Feature Distillation: Stabilizes knowledge transfer during pruning by aligning features across layers.
PocketSR achieves 146M parameters, processes 4K images in 0.8s, and matches SOTA single/multi-step methods in fidelity and perceptual quality.

**Questions:**

Please see Cons.

**Ethical Concerns:**

["NO or VERY MINOR ethics concerns only"]

**Final Justification:**

The rebuttal addressed some of my concerns. After reading the other reviewers' comments, I decided to raise the score to borderline accept.

**Limitations:**

Yes

**Quality:**

2

**Strengths And Weaknesses:**

Pros:

1.Novelty: The online annealing pruning and LiteED design offer fresh solutions to diffusion model compression. The dual-path feature injection effectively mitigates information bottlenecks.

2.Practical Impact: PocketSR’s efficiency (62.5 FPS for 512×512 images on A100) makes it viable for mobile/edge devices. The 10× parameter reduction over StableSR is compelling.

3.Rigorous Evaluation: Extensive comparisons on RealSR/DRealSR benchmarks show PocketSR outperforms SOTA in LPIPS, DISTS, and NIQE.


Cons：

1.Insufficient reproducibility of the experiment. The code is not open-source: the author declares that it cannot be open-sourced due to company policy (Page 15), which seriously affects reproducibility. It is recommended to provide pseudo-code or schematic diagrams of the inference process. And Section 4.1 does not adequately describe the training details.

2.Results Analysis: In Table 2 PocketSR has lower PSNR/SSIM than ResShift, but better LPIPS/NIQE. Discussion is needed on whether this trade-off affects specific application scenarios.

3.Writing. Section 3.1 “LiteAE” (Page 2&4) and “LiteED” (Abstract) are suspected to be mixed and need to be corrected. Comparative Clarity: Is the “Parameters (M)” line in Table 2 mislabeled, with 119M for both ResShift/SinSR?

4.Edge Deployment Realism: The paper overlooks comparisons with non-diffusion SR models (e.g., transformer-based and CNN-based), leaving the efficiency breakthrough incompletely contextualized. Also, can authors report on the inference speeds on common GPUs such as 3090 and 4090, since the authors claim it is for edge computing?

---

> ### Author Rebuttal · Authors · 2025-07-31
>
> We sincerely thank the reviewer for taking the time to read our paper and provide valuable feedback. We are pleased to see that the reviewer finds our work both novel and practically valuable. We deeply appreciate your insights, and in the following, we address each of your concerns carefully. All discussions and additional experiments presented in this rebuttal will be incorporated into the camera-ready version of the paper.
>
> ---
> > **[W1] Insufficient reproducibility.**
>
> We fully understand and agree with the reviewer’s concern. In response, we have decided to conduct a code reimplementation, within the boundaries of our company’s policy, and plan to release it publicly upon acceptance to better support the community.
>
> Regarding the training details mentioned by the reviewer, we would like to clarify that they are indeed provided in Section 3.3 rather than Section 4.1. Nonetheless, we will further enrich the description of training settings and implementation details in the camera-ready version to improve clarity and reproducibility.
>
> ---
> > **[W2] Performance trade-off (better LPIPS/NIPE but lower PSNR/SSIM than ResShift).**
>
> We sincerely thank the reviewer for the insightful and professional comments. We would like to clarify this concern from two perspectives:
>
> 1. PSNR and SSIM primarily reflect pixel-level similarity, which does not always align with human visual perception—a limitation that has been widely discussed in prior works on image super-resolution and quality assessment [1,2,3]. Moreover, it is possible for overly smoothed outputs to yield higher PSNR/SSIM scores. For example, we found that **simply upsampling the low-resolution images in DRealSR using bicubic interpolation achieves a PSNR of 30.58 and SSIM of 0.8312**—surpassing all super-resolution methods reported in Table 2. A similar phenomenon is observed on RealSR. This clearly contradicts the fundamental goal of super-resolution, which is to recover fine and realistic image details.
> 2. Given the limitations of PSNR/SSIM, we advocate for a multi-metric evaluation strategy rather than relying on a single indicator. From the perceptual similarity perspective, our method achieves the best performance in both LPIPS and DISTS (which measure different aspects from PSNR/SSIM), indicating that our results are most consistent with human perception. On the pixel-wise alignment side, while our method slightly lags behind ResShift and SinSR, it performs on par with previous multi-step diffusion-based methods such as StableSR and DiffBIR, as well as single-step approaches like OSEDiff and AdcSR—demonstrating comparable pixel-level fidelity. In terms of image detail and texture, our method achieves the best NIQE score and comparable MUSIQ, further reflecting its ability to produce high-quality outputs.
>
> In summary, our method delivers competitive or even superior performance across multiple dimensions, while being significantly more efficient in computation, highlighting its practical value on common application scenarios.
>
> [1] Scaling up to excellence: Practicing model scaling for photo-realistic image restoration in the wild, CVPR 2024
>
> [2] CoSeR: Bridging Image and Language for Cognitive Super-Resolution, CVPR 2024
>
> [3] Image quality assessment: Unifying structure and texture similarity, TPAMI 2020
>
> ---
> > **[W3] Writing and clarity.**
>
> We sincerely appreciate the reviewer’s careful examination. The mention of LiteAE was a typographical error, and we will correct it in the camera-ready version.
>
> Regarding the parameter counts reported in Table 2, we would like to clarify that the numbers are accurate. This is because SinSR is distilled from ResShift, and thus they share the same number of parameters. We respectfully invite the reviewer to refer to Table 2 in OSEDiff[4], where the reported values are consistent with ours.
>
> [4] One-Step Effective Diffusion Network for Real-World Image Super-Resolution, NeurIPS 2024
>
> ---
> > **[W4-1] Comparisons with non-diffusion SR models.**
>
> We sincerely thank the reviewer for the insightful and professional feedback. To address your concern, we have conducted additional comparisons on the DRealSR dataset with several representative non-diffusion-based methods, including a CNN-based method (DASR [5]) and a Transformer-based method (FeMaSR [6]). The comparison results are presented in the table below (FPS measured on an A100 server).
>
> Our method significantly outperforms the competing approaches in terms of perceptual quality metrics such as LPIPS, DISTS, NIQE, and MUSIQ. While DASR shows stronger results in PSNR, SSIM, and efficiency, we believe this is partly due to its extremely lightweight design, which—while advantageous in terms of speed and model size—also limits its ability to recover fine-grained details from complex real-world degradations, resulting in smoother outputs. As discussed in **[W2]**, such characteristics can sometimes lead to higher PSNR and SSIM scores that may not fully reflect perceptual quality.
>
> $$
> \begin{array}{l|cccccc|ccc}
> \hline
> \textbf{Methods} & \textbf{PSNR} \uparrow & \textbf{SSIM} \uparrow & \textbf{LPIPS} \downarrow & \textbf{DISTS} \downarrow & \textbf{NIQE} \downarrow & \textbf{MUSIQ} \uparrow & \textbf{Params (M)} & \textbf{MACs (G)} & \textbf{FPS} \\\\
> \hline
> \text{DASR[5] (CNN-based)} & \mathbf{29.75} & \mathbf{0.8262} & 0.3099 & 0.2275 & 7.586 & 42.41 & \mathbf{8} & \mathbf{46} & \mathbf{148.0} \\\\
> \text{FeMaSR[6] (Trans-based)} & 26.87 & 0.7569 & 0.3157 & 0.2239 & 5.910 & 53.71 & 34 & 384 & 16.8 \\\\
> \text{PocketSR (Ours)} & 28.05 & 0.7675 & \mathbf{0.2962} & \mathbf{0.2139} & \mathbf{5.809} & \mathbf{63.85} & 146 & 225 & 62.5 \\\\
> \hline
> \end{array}
> $$
>
> [5] Efficient and Degradation-Adaptive Network for Real-World Image Super-Resolution, ECCV 2022
>
> [6] Real-World Blind Super-Resolution via Feature Matching with Implicit High-Resolution Priors, ACMMM 2022
>
> ---
> > **[W4-2] Inference speed on common GPUs.**
>
> We sincerely thank the reviewer for the valuable suggestion. The table below presents the inference time of different methods on an RTX 4090 GPU. We observe that the inference speed on the 4090 is comparable to that on the A100, demonstrating that our method maintains high efficiency even when deployed on consumer-grade GPUs—highlighting its practicality for real-world applications.
> If the reviewer is interested in the performance of PocketSR in more constrained edge-computing environments such as mobile NPUs, we kindly refer you to our response to **Reviewer nGC4’s Weakness 2**. Specifically, PocketSR takes approximately 140ms to super-resolve a 512×512 image on a mobile NPU.
>
> $$
> \begin{array}{l|cccccc}
> \hline
> \textbf{4090 GPU} & \textbf{DASR} & \textbf{FeMaSR} & \textbf{SinSR} & \textbf{OSEDiff} & \textbf{AdcSR} & \textbf{PocketSR (Ours)} \\\\
> \hline
> \text{Inference Speed (ms)} & 7.23 & 59.83 & 129.05 & 112.70 & 41.67 & 16.78 \\\\
> \hline
> \end{array}
> $$

---

### Decision · Program_Chairs · 2025-09-17

**Decision:**

Accept (poster)

**Comment:**

Summary:
- This paper presents PocketSR, an ultra-lightweight, single-step generative model for real-world image super-resolution. It achieves state-of-the-art quality with only 146M parameters. It replaces the heavy VAE in Stable Diffusion with LiteED (97.5% fewer parameters), applies online annealing pruning to optimize the U-Net, and uses multi-layer feature distillation to preserve priors. PocketSR processes 4K images in 0.8s, matching or surpassing existing single- and multi-step methods while being practical for edge deployment.

Strength:
- Technical novelty: new approach for model compression using online annealing pruning and LiteED design.
- Practicality: efficient inference makes it practical for mobile and edge devices.

Weakness:
- The paper lacks clarity in several areas: the evaluation methodology and dataset details are unclear, the Stable Diffusion version used is unspecified, and the 4K SR quality analysis is insufficient.
- Reproducibility is limited by its decision not to open-source the model.

Overall, all reviewers are positive and appreciate the contributions preseted in the paper. The AC checks the reviews and agrees with the concensus.